# Design and Analysis of Three-Stage Amplifier for Driving pF-to-nF Capacitive Load Based on Local *Q*-Factor Control and Cascode Miller Compensation Techniques

**Qi Cheng [1,2], Weimin Li [1], Xian Tang [3] and Jianping Guo [1,4,*]**

[1]  School of Electronics and Information Technology, Sun Yat-sen University, Guangzhou 510006, China; Qi.Cheng@utdallas.edu (Q.C.); liwm6@mail2.sysu.edu.cn (W.L.)
[2]  School of Engineering and Computer Science, The University of Texas at Dallas, Richardson, TX 75080, USA
[3]  Graduate School at Shenzhen, Tsinghua University, Shenzhen 518060, China; tang.xian@sz.tsinghua.edu.cn
[4]  Silicon (Shenzhen) Electronic Technology Co., Ltd., Shenzhen 518000, China
[*]  Correspondence: guojp3@mail.sysu.edu.cn; Tel.: +86-20-8411-4462

**Abstract:** This paper presents a new frequency compensation approach for three-stage amplifiers driving a pF-to-nF capacitive load. Thanks to the cascode Miller compensation, the non-dominant complex pole frequency is extended effectively, and the physical size of the compensation capacitors is also reduced. A local *Q*-factor control (LQC) loop is introduced to alter the *Q*-factor adaptively when loading capacitance $C_L$ varies significantly. This LQC loop decides how much damping current should be injected into the corresponding parasitic node to control the *Q*-factor of the complex-pole pair, which affects the frequency peak at the gain plot and the settling time of the proposed amplifier in the closed-loop step response. Additionally, a left-half-plane (LHP) zero is created to increase the phase margin and a feed-forward transconductance stage is paralleled to improve the slew rate (SR). Simulated in 0.13-μm CMOS technology, the amplifier is verified to handle a 4-pF-to-1.5-nF (375× drivability) capacitive load with at least 0.88-MHz gain-bandwidth (GBW) product and 42.3° phase margin (PM), while consuming 24.0-μW quiescent power at 1.0-V nominal supply voltage.

**Keywords:** three-stage CMOS amplifiers; cascode miller compensation; local *Q*-factor control; pole-zero cancellation; wide drivability range

## 1. Introduction

The single-stage amplifier used to be one of the strongest candidates for precise analog signal processing when old CMOS technologies were employed because of its high-speed and inherent good stability characteristics. As proved in [1], a single-stage telescopic cascode amplifier can achieve up to 100-dB low-frequency voltage gain with 0.8-μm CMOS technology. However, as the output impedance of the MOSFET is further decreased due to the channel-modulation effect in the modern advanced CMOS technology, it is more difficult for traditional single-stage amplifiers to obtain high voltage gain. In that case, some techniques have been proposed to enlarge the voltage gain of single-stage amplifiers, such as output resistance boosting, transconductance ($G_m$) boosting and multiple small-gain stages cascading [2–4]. In these strategies, stability, output swing and power efficiency are always traded for voltage gain. More importantly, most of them cannot deliver the required high voltage gain (>100 dB) for the high accuracy applications requiring precision buffering.

Cascading multiple gain stages is a good way to get high voltage gain because it is potentially power-efficient with low supply voltage. One of the essential problems for multistage amplifiers is the

closed-loop stability. Generally speaking, there are at least three poles that exist in the transfer function of the loop gain of a three-stage amplifier. If the poles and zeros were distributed inappropriately, the multistage amplifier would encounter a closed-loop stability issue [5].

As to the stability criteria, it normally can be indicated by the parameters of phase margin (PM) or gain margin (GM) in the Bode plot for the design of single-stage and two-stage amplifiers. However, the stability analysis of multistage amplifiers is more complex than single- or two-stage amplifiers due to the existence of complex poles in high-order transfer functions [6]. Moreover, the key specifications (e.g., gain-bandwidth (GBW), PM, GM) are normally tied to the frequency compensation approach and the value of the capacitive load $C_L$.

Several compensation schemes for three-stage amplifiers have been reported in the past few decades [7–19]. Nested-Miller compensation (NMC) is known as one of the most classical pole-splitting techniques for three-stage amplifiers frequency compensation. The basic idea of NMC scheme is to capacitively nest several pairs of gain stages to achieve pole-splitting [7]. However, the bandwidth reduction, which is mainly caused by the required large value Miller capacitor, degrades the benefits of the technique. To tackle this problem, other compensation schemes based on NMC have been proposed [8–14], some of which could enlarge the GBW tenfold comparing with the traditional NMC technique. Generally, they either removed the inner Miller capacitor or replaced the outer compensation loop with more advanced compensation techniques [11,15–17] to extend complex-pole frequency $\omega_o$. In some others designs, like [20] and [21], either an active zero or a wide-bandwidth scalar is embedded in the multistage amplifier to extend the non-dominant pole frequency for driving an extremely large capacitive load. Naturally, these techniques can achieve better small-signal performance by increasing the product of load capacitor value and unit-gain frequency. These techniques, however, fail to tackle the problem of frequency peak at gain plot due to a large $Q$-factor of complex-pole when the load capacitance is dropped significantly [22]. As a result, in the transient step response, a high-frequency oscillation would appear and last for a long period [17].

Most existing frequency compensation schemes for three-stage amplifiers focus on maximizing the performance for a single value of capacitive load $C_L$ (especially the large $C_L$) to achieve better figure-of-merit (FOM) rather than extending the drivability range of $C_L$. However, the load capacitance can change in the range of pF–nF depending on applications such as headphone, liquid-crystal display (LCD) or microelectromechanical systems (MEMS) capacitive sensors [23–25]. In other words, an amplifier with wide capacitive loading drivability can find more applications and is easy to be reused in a different environment. As a result, there is no need to design the amplifier circuits case by case when the loading capacitance is different, which is helpful to shorten the design procedure and save the production cost. The technique for extending the drivability range of two-stage amplifiers has been studied in [26]. Comparing with the two-stage amplifiers, it is more difficult to stabilize and even more challenging to extend the drivability range for three-stage amplifiers. Although some three-stage amplifiers with wide bandwidth have been reported to have large driving capability for large capacitive load [21,27], it is hard to find amplifier designs able to combine the possibility to drive capacitive load in the pF and nF range with low quiescent power and small active area [28–30].

Expanding the report in [31], this paper provides the analysis and design insights for a low-power three-stage amplifier capable of driving the pF-to-nF capacitive load. The cascode Miller compensation in the outer feedback loop helps to extend the non-dominant complex-pole frequency and the physical size of the compensation capacitors is reduced as well. The $Q$-factor of the complex-pole pair is controlled by the local feedback loop adaptively, which improves the frequency response and shortens the transient settling time. In this design, 375× capacitive load drivability is realized for the proposed amplifier. Additionally, at least 0.88-MHz GBW and 0.41-V/µs average slew rate (SR) of the proposed three-stage amplifiers are achieved with 24.0-µW power consumption.

The rest of this paper is organized as follows. In Section 2, several previous advanced frequency compensation techniques for three-stage amplifiers are reviewed. The pole-zero locus of three-stage amplifiers with wide load variations is investigated. In Section 3, the proposed frequency compensation

approach with local *Q*-factor control is presented. The transfer function, stability criteria, and transient response are also addressed. In Sections 4 and 5, the circuit implementation of the proposed topology, simulation results and corresponding discussions are given. In Section 6, we conclude the performance of the proposed design and its advantages.

## 2. Review of Previous Frequency Compensation Techniques under Large Load Variations

### 2.1. Nested Miller Compensation (NMC)

Figure 1a,b shows the classical NMC topology and the pole-zero locus under 100 times $C_L$ variation, respectively. The complex-pole in NMC amplifier is given by

$$\omega_{o(NMC)} = \sqrt{\frac{G_{m2}G_{mL}}{C_{m2}C_L}},\tag{1}$$

and the relevant *Q*-factor is given by

$$Q_{(NMC)} = \frac{1}{G_{mL} - G_{m2}} \sqrt{\frac{G_{m2}G_{mL}C_L}{C_{m2}}},\tag{2}$$

where the parameters are identified in Figure 1a. As indicated by Equation (1), the complex-pole frequency $\omega_o$ is related to the Miller capacitor $C_{m2}$ of the inner feedback loop. Therefore, it is natural to reduce $C_{m2}$ to achieve a higher $\omega_o$ and thus a higher GBW. However, $C_{m2}$ value is related to the *Q*-factor which is presented in Equation (2). In order to extend $\omega_o$ while suppressing *Q*-factor, the only way is to enlarge $G_{m2}$ and $G_{mL}$ which inevitably increases power consumption.

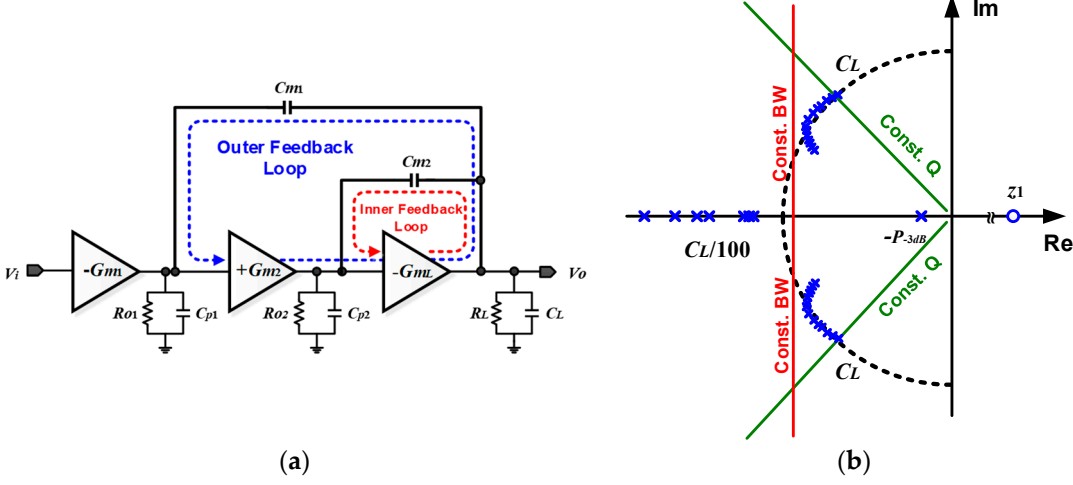

**(a)**                                                                     **(b)**

**Figure 1.** Three-stage nested-Miller compensation (NMC) amplifier: (**a**) topology, and (**b**) the pole-zeros locus under 100× $C_L$ variation (unscaled).

To achieve the maximum flat frequency response, for a three-stage NMC amplifier, it turns out that the bandwidth is degraded by 75%, by comparing with a single-stage amplifier [8]. This weakens the advantages of three-stage amplifiers over single-stage amplifiers. Furthermore, NMC amplifiers are difficult to drive a nano-Farads large capacitive load, which will deteriorate the pole-splitting effect caused by the floating Miller capacitor. More power could be consumed to create high-frequency non-dominant poles under large $C_L$ condition.

When dealing with large load variations, we assume the NMC amplifier is designed for Butterworth poles constellation. As the load capacitance $C_L$ is reduced, the $Q_{(NMC)}$ will decrease because it is proportional to $\sqrt{C_L}$ which can be seen from Equation (2). The unity-gain-frequency will only be limited by the first non-dominant pole frequency. Thus, good stability of NMC amplifier under a

small capacitive load can be easily achieved. The pole-zeros locus under load variations from $C_L$ to $C_L/100$ of the NMC topology is shown in Figure 1b. As shown in this Figure, with the $C_L$ decreasing, the poles move so that complex-pole frequency $\omega_o$ increases, but $Q_{(NMC)}$ is constrained to a narrow range. Eventually, $Q$ will be reduced to 0.5 and the complex-pole will split into two real poles. One pole moves to a higher frequency, the other one goes lower.

### 2.2. Damping Factor Control Frequency Compensation (DFCFC)

The damping factor control frequency compensation (DFCFC) technique presented in [10] aims to reduce the static power under large capacitive loading conditions. A circuit block for controlling the damping factor $\zeta$ (= 1/2$Q$), composed of a $G_m$ cell in parallel with a Miller capacitor $C_{m2}$, is adopted to increase the $\zeta$ of the non-dominant complex poles and stabilize the multi-stage amplifier.

The complex-pole in DFCFC amplifier is given by

$$\omega_{o(DFCFC)} = \sqrt{\frac{(G_{m2}G_{mL} + G_{mf}G_{m4})}{C_{p2}C_L}}, \tag{3}$$

and the relevant $Q$-factor is given by

$$Q_{(DFCFC)} = \frac{1}{G_{m4}}\sqrt{\frac{(G_{m2}G_{mL} + G_{mf}G_{m4})C_{p2}}{C_L}}, \tag{4}$$

where the parameters are identified in Figure 2a. Comparing Equation (3) to Equation (1), it is easy to find that DFCFC amplifiers can achieve wider bandwidth by a factor of $\sqrt{C_{m2}/C_{p2}}$, which is larger than 1 since $C_{p2}$ is the parasitic capacitance and is often less than $C_{m2}$. According to [10], the $Q_{(DFCFC)}$ can be suppressed by injecting more damping current into damping factor control (DFC) block. This can be realized by reducing the output impedance of the second gain stage ($v_1$ in Figure 2a) in the high-frequency range. The equivalent impedance looking into DFC block is described by

$$Z_{eq(DFC)} = \frac{sC_{p4}R_{o4} + 1}{s^2C_{m2}C_{p4}R_{o4} + sC_{m2}(1 - G_{m4}R_{o4})}. \tag{5}$$

From Equation (5), $Z_{eq(DFC)}$ is an increasing function with $G_{m4}$, so that a smaller $G_{m4}$ will result in a smaller $Z_{eq(DFC)}$ and thus the DFC block only consumes a small amount of power.

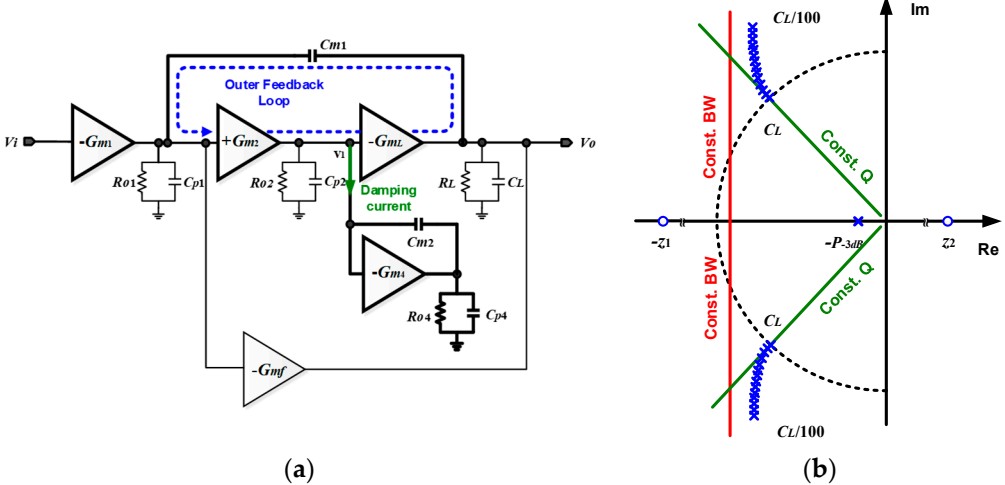

(**a**)　　　　　　　　　　　　　　　　　　　　　　　(**b**)

**Figure 2.** Three-stage damping factor control frequency compensation (DFCFC) amplifier: (**a**) topology, and (**b**) the pole-zeros locus under 100× $C_L$ variation (unscaled).

As to load variations, $Q_{(DFCFC)}$ is proportional to $1/\sqrt{C_L}$ which can be seen from Equation (4). Even though, the $Q$-factor of DFCFC amplifiers can be adjusted by $G_{m4}$ from DFC block. Once the $G_{m4}$ is decided, the $Q_{(DFCFC)}$ will still increase by 10 times when $C_L$ drops to $C_L/100$ according to Equation (4). If $C_L$ decreases further, the complex poles would exhibit a higher $Q$-factor which causes an unsettled system in closed-loop step response. In fact, it is mentioned in [10] that the DFCFC scheme is effective only when driving the large capacitive load.

### 2.3. Cascode Miller Compensation with Local Impedance Attenuation (CLIA)

Another technique known as cascode Miller compensation with local impedance attenuation (CLIA) to control the $Q$-factor is presented in [32]. A passive RC-series network is added to stabilize the amplifier by attenuating the small-signal output impedance of the second stage in the high-frequency range.

The complex-pole in CLIA amplifier is given by

$$\omega_{o(CLIA)} = \sqrt{\frac{G_{m2}G_{mL}G_{mc}R_a}{C_{p1}C_L}},\tag{6}$$

and the relevant $Q$-factor is given by

$$Q_{(CLIA)} = C_{m1}\sqrt{\frac{g_{m2}g_{mL}R_a}{g_{mc}C_{p1}C_L}},\tag{7}$$

where the parameters are identified in Figure 3a. With the advantage of the cascode compensation [33], the complex pole frequency $\omega_o$ is pushed to a higher frequency by a factor of approximately $\sqrt{G_{mc}R_a}$ than the topologies using simple Miller compensation at the outer feedback loop, like DFCFC. As indicated by Equations (6) and (7), CLIA amplifiers achieve higher bandwidth than NMC amplifiers. Additionally, the $Q$-factor of the CLIA amplifier can be adjusted by setting appropriate values of $R_a$ to define the high-frequency equivalent impedance at the output node at the second gain stage ($v_2$ in Figure 3a) as

$$Z_{eq(LIA)} = R_a + \frac{1}{sC_a}\tag{8}$$

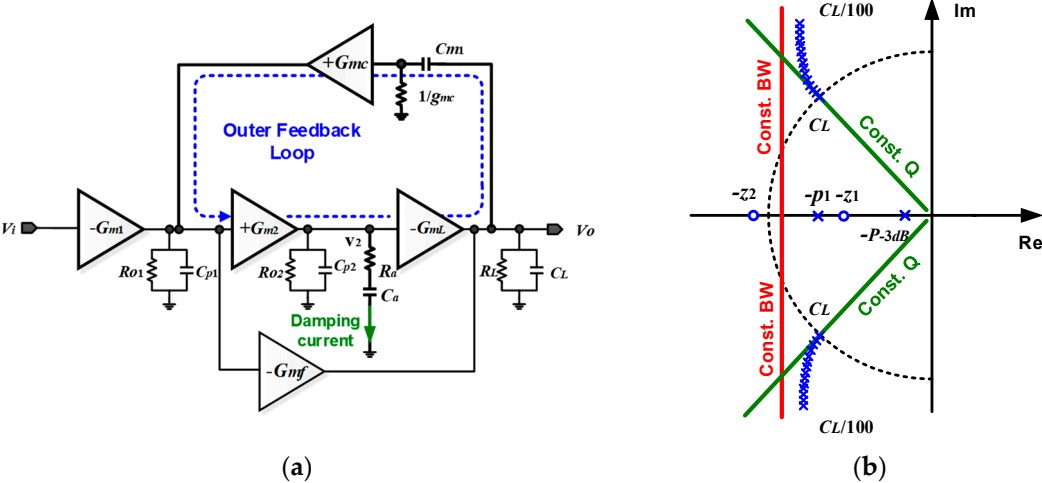

(**a**)　　　　　　　　　　　　　　　　　　　　　　(**b**)

**Figure 3.** Three-stage cascode Miller compensation with local impedance attenuation (CLIA) amplifier: (**a**) topology, and (**b**) the pole-zeros locus under 100× $C_L$ variation (unscaled).

From Equation (8), $Z_{eq(LIA)}$ is an increasing function with $R_a$, so that the LIA block absorbs more damping current when $R_a$ is reduced and then the $Q_{(CLIA)}$ decrease, which can be proven by Equation (7).

As to load variations, $Q_{(CLIA)}$ is proportional to $1/\sqrt{C_L}$ which can be seen from Equation (7). Similar to the DFCFC scheme, the LIA block helps optimize $Q$-factor atfixed load capacitance. When $C_L$ drops to $C_L/100$, the $Q_{(CLIA)}$ will increase by 10 times. If $C_L$ decreases further, the complex poles would exhibit a larger $Q$-factor.

## 3. Proposed Cascode Miller-Compensation with Local $Q$-Factor Control (CLQC)

As mentioned in previous sections, a high $Q$-factor could result in unstable amplifiers if $C_L$ is reduced or increased significantly according to different design topologies. The idea of the proposed work is to design an advanced compensation topology that can control the $Q$-factor of the complex pair in a proper range when $C_L$ changes significantly.

### 3.1. Structure

Figure 4 shows the equivalent diagram of the proposed three-stage cascode Miller-compensation with local $Q$-factor control (CLQC) amplifier [31]. It consists of two inverting gain stages, a non-inverting gain stage, two current buffered Miller compensation blocks, and one feed-forward block. Like [16], the cascode Miller compensation block ($+G_{ma1}$ and $C_{m1}$) eliminates the feed-forward signal path (which may cause the right-half-plane (RHP) zero) that exists in simple Miller compensation, creates an LHP (left half-plane) zero and extends the complex-pole frequency. A feed-forward path ($G_{mf}$) is added to form a push-pull output stage with $G_{mL}$ to improve the transient performance. Unlike the realization in [15], the other local Miller compensation block ($-G_{ma2}$ and $C_{m2}$) is not aimed at creating an LHP zero for pole-zero cancellation but composing a local $Q$-factor control loop with the second gain stage. It controls the amount of damping current to be injected in $C_{m2}$ to alter the small-signal impedance at the output node of $G_{m2}$ ($v_3$ in Figure 4), which affects the $Q$-factor of the corresponding complex-pole.

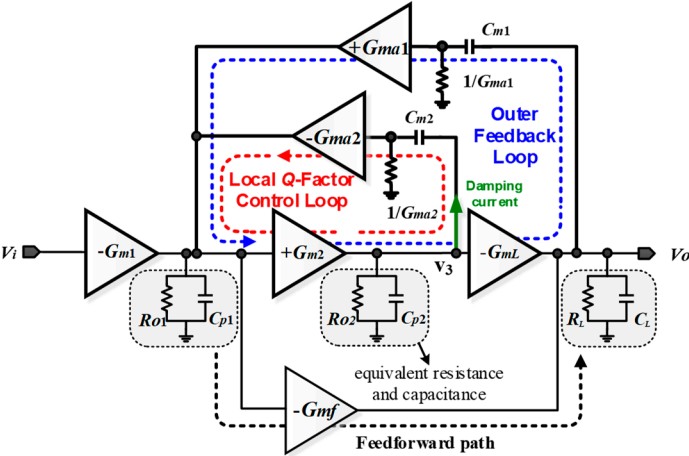

**Figure 4.** Equivalent diagram of the proposed three-stage cascode Miller-compensation with local $Q$-factor control (CLQC) amplifier.

### 3.2. Small-Signal Analysis of the Proposed Three-Stage CLQC Amplifier

The equivalent small-signal model of the proposed three-stage CLQC amplifier is shown in Figure 5, where $G_{mi}$, $R_{oi}$, and $C_{pi}$ are noted as the equivalent transconductance, output resistance and the lumped capacitance at the $i^{th}$ gain stage, $G_{ma1}$ and $G_{ma2}$ are the equivalent transconductances of the current buffered Miller compensation stages, and $G_{mf}$ is the feed-forward transconductance. In this model, the output parasitic capacitance is lumped into the load capacitor $C_L$.

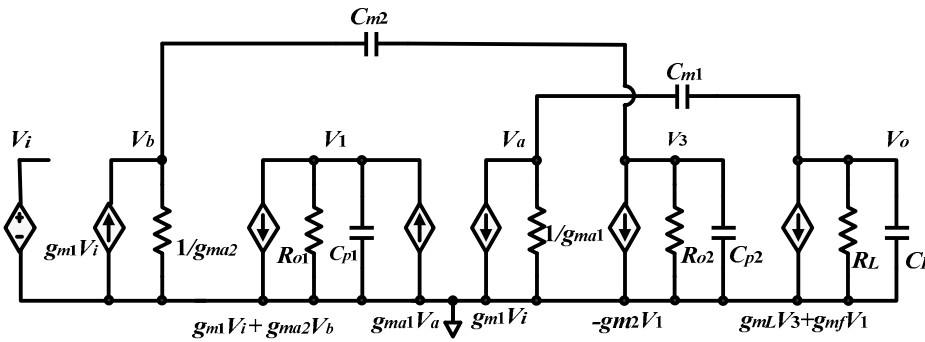

**Figure 5.** The equivalent small-signal model of the proposed three-stage CLQC amplifier.

To analyze the stability of the proposed amplifier, the following common assumptions are made to simplify the transfer function [11].

$$G_{m1}R_{o1}, G_{m2}R_{o2}, G_{mL}R_L >> 1, R_a = 1/G_{ma}, G_{mf} >> G_{ma2}, \text{ and } C_L >> C_{m2} \text{ and } C_{m1} >> C_{p1}, C_{p2}. \quad (9)$$

The overall transfer function $A_v(s)$ of the proposed amplifier is presented as

$$A_v(s) = \frac{A_{dc}(1+b_1s+b_2s^2+b_3s^3+b_4s^4)}{(1+\frac{s}{P_{-3dB}})(1+a_2s+a_3s^2+a_4s^3+a_5s^4)} = \frac{G_{m1}G_{m2}G_{mL}R_{o1}R_{o2}R_L(1+b_1s+b_2s^2+b_3s^3+b_4s^4)}{(1+sC_{m1}G_{m2}G_{mL}R_{o1}R_{o2}R_L)(1+a_2s+a_3s^2+a_4s^3+a_5s^4)}, \quad (10)$$

where terms of denominator and numerator are defined as

$$a_2 = \frac{C_{m2}[(C_L+C_{m2})G_{m2}+C_{m1}G_{m2}G_{mL}(1/G_{ma2})+C_{m1}G_{mf}]}{C_{m1}G_{m2}G_{mL}}, a_3 = \frac{C_{m2}C_L}{G_{ma1}G_{mL}}, a_4 = \frac{C_{p1}C_LC_{m2}}{G_{ma1}G_{m2}G_{mL}}, a_5 = \frac{C_{p1}C_{p2}C_LC_{m2}}{G_{m2}G_{mL}G_{ma1}G_{ma2}},$$

$$b_1 = \frac{C_{m2}}{2G_{ma2}} + \frac{C_{m1}}{2G_{ma1}} + \frac{C_{m2}G_{mf}}{G_{m2}G_{mL}}, b_2 = \frac{C_{m1}C_{m2}(G_{mf}-G_{m2})}{2G_{m2}G_{mL}G_{ma1}}, b_3 = \frac{-C_{p1}C_{m1}C_{m2}}{2G_{m2}G_{ma1}G_{ma2}}, b_4 = \frac{-C_{p1}C_{p2}C_{m1}C_{m2}}{2G_{m2}G_{mL}G_{ma1}G_{ma2}}. \quad (11)$$

From Equation (10), it can be found $A_{dc} = G_{m1}G_{m2}G_{mL}R_{o1}R_{o2}R_L$ is the DC voltage gain, and the dominant pole $P_{-3dB}$ is $1/C_{m1}G_{m2}G_{mL}R_{o1}R_{o2}R_L$. Note that this is a very general transfer function and the further approximation will be analyzed in the following section.

### 3.3. Stability Analysis Under Large $C_L$ Variation

As mentioned earlier, the complex-pole frequency $\omega_o$ and Q-factor will change according to different loading capacitance $C_L$. In order to analyze the functionality of the proposed compensation scheme that can handle a wide range of capacitive loads, the transfer functions of amplifiers with different $C_L$ should be studied in different cases.

*Case I*: When $C_L$ is large (nano-Farads level), the non-dominant poles are separated into few real ones, the poles and zeros which locate at low-frequency dominate the frequency response. It turns out the amplifier can be approximated by a two-pole system. In this case, the gain transfer function $A_v(s)$ can be estimated as

$$A_v(s) = \frac{A_{dc}}{(1+\frac{s}{\omega_{-3dB}})(1+\frac{s}{\omega_{p1}})} \approx \frac{1}{\frac{s}{GBW}(1+\frac{s}{\frac{C_{m1}G_{mL}}{C_LC_{m2}}})}, \quad (12)$$

and the corresponding phase margin is given by

$$PM \approx 90^o - \phi(\omega_{p1}) = 90^o - \tan^{-1}(\frac{GBW}{\omega_{p1}}), \quad (13)$$

where $\omega_{-3dB}$ and $\omega_{p1}$ are the dominant and non-dominant poles of the amplifier, and the $GBW = G_{m1}/C_{m1}$ is the gain bandwidth product which should always be smaller than $\omega_{p1}$ to ensure good stability. As the load capacitor $C_L$ increases, $\omega_{p1}$ will move towards low frequency and thus degrades phase margin.

According to Equations (12) and (13), the maximum load capacitance $C_{L\_max}$ is decided by the minimum phase margin $PM\_min$, which can be calculated as

$$C_{L\_max} = \frac{C_{m1}{}^2 G_{mL} \cot(PM\_min)}{G_{m1} C_{m2}} \tag{14}$$

*Case II*: When $C_L$ is moderate (hundred pico-Farads level), non-dominant pole $p_1$ moves towards high frequency and merges another pole $p_2$ to a complex-pole pair $p_{1,2}$. Meanwhile, an LHP zero $z_1$ generated by $G_{ma1}$ and $C_{m1}$ can increase the phase margin. The gain transfer function $A_v(s)$ is simplified as

$$A_v(s) = \frac{A_{dc}(1 + \frac{s}{\frac{2G_{ma1}}{C_{m1}}})}{(1 + \frac{s}{\omega_{-3dB}})(1 + \frac{1}{Q}\frac{1}{\omega_{p1,2}}s + \frac{1}{\omega_{p1,2}^2}s^2)}, \tag{15}$$

where the frequency of the complex-pole $p_{1,2}$ is given by

$$\omega_{o(CLQC)} = \omega_{p1,2} = \sqrt{\frac{G_{ma1}G_{mL}}{C_{m2}C_L}}, \tag{16}$$

and correspondingly its $Q$-factor is expressed as

$$Q_{(CLQC)} = \sqrt{\frac{C_L G_{mL}}{G_{ma1} C_{m2}}} \cdot \frac{C_{m1} G_{m2}}{(C_L + C_{m2})G_{m2} + C_{m1} G_{m2} G_{mL}(1/G_{ma2}) + C_{m1} G_{mf}} = \frac{k_1}{a \sqrt{C_L} + b/\sqrt{C_L}}. \tag{17}$$

We note that in Equation (17), $k_1 = C_{m1}\sqrt{\frac{G_{mL}}{G_{ma1}C_{m2}}}$, $a = 1$, and $b = C_{m2} + C_{m1}G_{mL}(1/G_{ma2}) + C_{m1}G_{mf}(1/G_{m2})$.

It is obvious that the frequency of the complex poles $\omega_o$ is a decreasing function with the loading capacitance $C_L$, which indicates the non-dominant poles are away from the unity-gain frequency (UGF) and will not cause a stability issue. The major challenge becomes to satisfy the conflicting requirements of $Q$-factor of complex poles at either light or heavy $C_L$ condition because the $Q$-factor changes with the variation of loading capacitor $C_L$. In fact, a high $Q$-factor exhibits a gain peak in magnitude plot and low $Q$-factor results in two separated real poles, which is not an optimized solution for better achievable bandwidth.

In order to avoid the obvious frequency peak showing at magnitude Bode plot of the amplifier, a high $Q$-factor of complex poles is unwanted. From Equation (17), to suppress the $Q$-factor, we either need to reduce $G_{ma2}$, $C_{m1}$ or increase $G_{mf}$ and $C_{m2}$. Unfortunately, most of the circuit parameters are interrelated. It is difficult to adjust them independently for the optimization of the frequency response [13]. For instance, a larger $C_{m2}$ decreases the $Q$-factor, but the complex-pole would also be removed to a lower frequency in that case, as indicated in Equation (16).

Even though, the mathematical expression at least gives an intuitive insight to control the $Q$-factor and natural frequency of the complex poles. The bellowing expression can be found from (17)

$$Q_{max} = \frac{k_1}{2\sqrt{ab}}. \tag{18}$$

The relationship between $Q$-factor and gain peak in the Bode plot has been studied in [15]. In many three-stage amplifier designs [1–10], for their non-dominant complex-pole, the relevant $Q$-factor is normally set to be $1/\sqrt{2}$ to make the amplifiers feature with third-order Butterworth frequency response when they are configured as a unity-feedback system. However, the Q-factor is always changed according to different $C_L$ and thus $1/\sqrt{2}$ is the least value for $Q_{max}$. On the other hand, to achieve good frequency response under wide output capacitance range, $Q_{max}$ should be smaller

than 2 to avoid the obvious frequency peak in the open-loop magnitude plot. Therefore, in this design, $1/\sqrt{2} < Q_{max} < 2$. Subsequently, the phase margin with pole-zero cancellation is given by

$$PM = 90^o - \phi(\omega_{p1,2}) + \phi(z_1) = 90^o - \tan^{-1}\left[\frac{\frac{GBW}{\omega_o}}{Q(1 - (\frac{GBW}{\omega_o})^2)}\right] + \tan^{-1}\left(\frac{GBW}{z_1}\right). \tag{19}$$

*Case III*: When $C_L$ is very small (pico-Farads level), the non-dominant complex-pole pair related to $C_L$ will locate at very high frequency. Even though, the zeros and poles (or complex poles) located at high frequency can still affect the stability. In this case, the transfer function $A_v(s)$ should be studied as

$$A_v(s) = \frac{A_{dc}(1 + b_1 s)(1 + b_2 s)}{(1 + \frac{s}{p_{-3dB}})(1 + \frac{1}{Q_o}\frac{1}{\omega_o}s + \frac{1}{\omega_o^2}s^2)(1 + \frac{1}{Q_1}\frac{1}{\omega_1}s + \frac{1}{\omega_1^2}s^2)}. \tag{20}$$

When $C_L$ becomes extremely small, the high-frequency complex pair $p_{3,4}$ can move to right-half-plane (RHP). However, it is known that RHP pole causes unstable negative feedback system [34]. Therefore, the *Routh–Hurwitz* stability criterion which has been widely used in multistage amplifiers design can be used to help decide design parameters, including the minimum loading capacitance. The *Routh–Hurwitz* stability criterion was simply evaluated by constructing Table 1. To achieve good stability for the proposed three-stage amplifier, all coefficients must be larger than zero.

**Table 1.** Routh parameter expansion for the 4th order polynomial of the Equation (10).

| Coefficients | Expansion |
|---|---|
| $A_0$ | 1 |
| $A_1$ | $\frac{C_{m2}[(C_L + C_{m2})G_{m2} + C_{m1}G_{m2}G_{mL}(1/G_{ma2}) + C_{m1}G_{mf}]}{C_{m1}G_{m2}G_{mL}}$ |
| $A_2$ | $\frac{C_{m2}C_L}{G_{ma1}G_{mL}}$ |
| $A_3$ | $\frac{C_{p1}C_L C_{m2}}{G_{ma1}G_{m2}G_{mL}}$ |
| $A_4$ | $\frac{C_{p1}C_{p2}C_L C_{m2}}{G_{m2}G_{mL}G_{ma1}G_{ma2}}$ |
| $B_1 = (A_3 A_2 - A_4 A_1)/A_3$ | $\frac{C_L C_{m2}}{G_{mL}G_{ma1}} - \frac{C_{p2}}{G_{ma2}}A_1$ |
| $B_2 = A_0$ | 1 |
| $C_1 = (B_1 A_1 - A_3 A_0)/B_1$ | $A_1 - \frac{A_3}{B_1}$ |
| $D_1 = A_0$ | 1 |

According to Table 1, the minimum load capacitance $C_L$ will be decided by the boundary condition of coefficient $B_1$ and $C_1$ being positive, which can be calculated as

$$C_{L\_min} = min\{B_1(C_L) > 0, \ C_1(C_L) > 0\}. \tag{21}$$

The phase margin with pole-zero cancellation is given by

$$\begin{aligned} PM &= 90^o - \phi(\omega_{p1,2}) + \phi(z_1) - \phi(\omega_{p3,4}) + \phi(z_2) \\ &= 90^o - \tan^{-1}\left(\frac{(\frac{GBW}{\omega_o})}{Q_o(1 - (\frac{GBW}{\omega_o})^2)}\right) + \tan^{-1}\left(\frac{GBW}{z_1}\right) - \tan^{-1}\left(\frac{(\frac{GBW}{\omega_1})}{Q_1(1 - (\frac{GBW}{\omega_1})^2)}\right) + \tan^{-1}\left(\frac{GBW}{z_2}\right). \end{aligned} \tag{22}$$

It is worth mentioning that if the circuit parameters of the amplifier have been set carefully to make sure all of the coefficients in Table 1 to be positive when $C_L$ is zero, then the ideally good stability under no load configuration of the proposed amplifier could be achieved.

### 3.4. Benefits of the Local Q-Factor Control (LQC) Loop and Cascode Miller Compensation

To better understand the main contribution of the proposed Local *Q*-Factor Control (LQC) loop, Figure 6 is given to illustrate the pole locus of the proposed amplifier under 1000x $C_L$ variations. Like the NMC scheme, when $C_L$ is reduced, the *Q*-factor of complex-pole is modified to a range instead of being proportional or inversely proportional to $\sqrt{C_L}$. This is because the local *Q*-factor control circuit generates necessary damping current under wide-range $C_L$. Unlike the NMC scheme, the proposed amplifier with LQC block requires a very small value $C_{m2}$ to deal with the tradeoff between $\omega_o$ and the *Q*-factor. On the other hand, the transconductance stage $G_{ma2}$ for composing a local feedback loop can be embedded into the first gain stage. In that case, there is no extra quiescent current in the LQC circuit, and the figure of the merit of the amplifier can be improved.

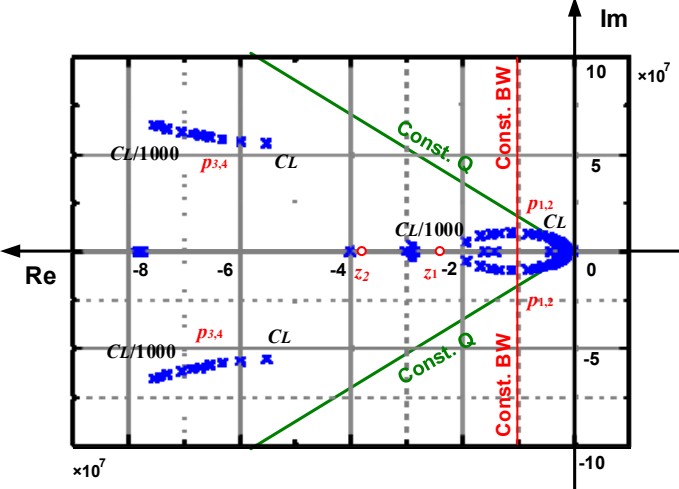

**Figure 6.** Pole locus of the proposed amplifier when the load capacitance varies from $C_L$ to $C_L/1000$ (scaled).

To further demonstrate the benefits of the proposed scheme over NMC scheme and simple cascode Miller compensation, all cases are simulated with same devices parameters except that of the compensation capacitor as noted in Figure 7, where the frequency responses of NMC, cascode Miller compensation, and the proposed design when driving a 400-pF capacitive load are shown. According to Figure 7, the structures applying cascode Miller compensation give over 60 times GBW than that of NMC structure due to their high-frequency complex poles. However, lacking the LQC circuit for damping current control, the complex poles in the simple cascode Miller compensated amplifier exhibit a high *Q*-factor which causes a gain peak. With the help of the LQC circuit, the *Q*-factor can be adjusted. It can be found in Figure 7 that there is no gain peaking in the proposed design.

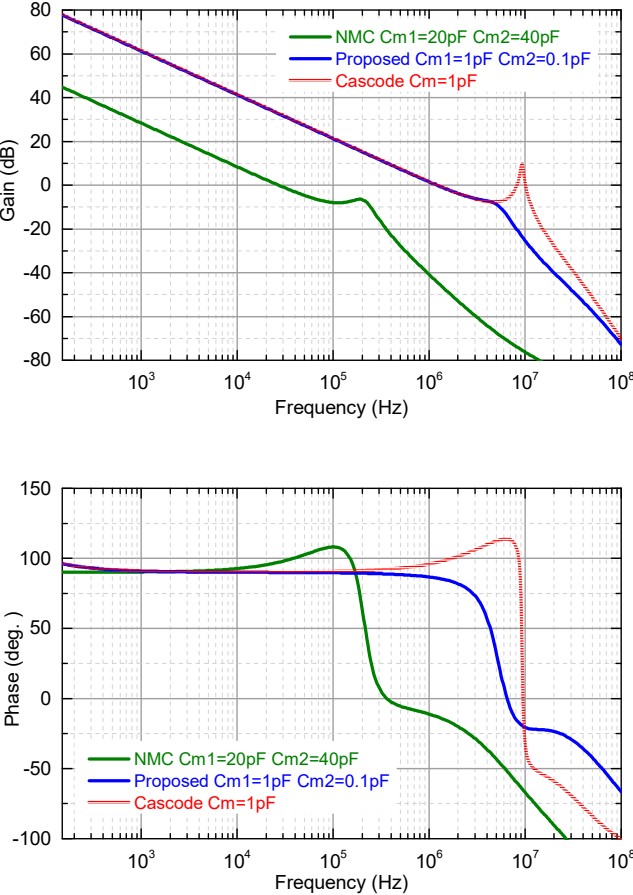

**Figure 7.** Frequency responses of NMC, cascode compensation without and with Local *Q*-Factor Control (LQC).

## 4. Circuit Implementation

The schematic of the proposed three-stage amplifier with CLQC technique is depicted in Figure 8 [31]. The 1st stage uses a folded cascode structure (M$_1$–M$_9$). The 2nd stage adopts a current mirror to form the non-inverting stage (M$_{10}$–M$_{14}$). Assuming the transconductance of M$_{11}$, M$_{12}$ and M$_{13}$ are $g_{m2,1}$, $g_{m2,2}$ and $g_{m2,3}$, respectively, the overall transconductance $g_{m2}$ is, therefore $(g_{m2,1}*g_{m2,3})/g_{m2,2}$. The 3$^{rd}$ stage is formed by a common-source stage (M$_{16}$). The cascode Miller compensation is realized by $C_{m1}$ and the common-gate stage (M$_7$). The local *Q*-factor control block contains Miller capacitor $C_{m2}$ and the common-source stage (M$_9$). The NMOS transistor M$_{15}$ generates a feed-forward path, which is helpful to enhance the transient performance. The quiescent current of the amplifier core circuit for each branch is labeled properly in Figure 8.

As to optimize the compensation capacitor value of the proposed CLQC amplifier, the value of $C_{m1}$ and $C_{m2}$ are set to be 1 and 0.05 pF, respectively, for the extreme capacitive load ranging from 4 pF to 1.5 nF. The values of compensation capacitors are obtained from the analysis in the previous section. The value of $C_{m1}$ is optimized according to the tradeoff between PM and GBW from Equations (12) and (13). The $C_{m2}$ value is obtained from the worst-case *Q*-factor ($Q_{(CLQC)\_max}$) value according to Equations (17) and (18). The $Q_{(CLQC)\_max}$ is obtained from the maximum gain peak magnitude of 20log(*Q*) caused by the complex pole. In this design, the maximum gain peak is suppressed to 3.5 dB which is about 10 dB smaller than the worst-case gain margin and the corresponding $Q_{(CLQC)\_max}$ value is 1.5.

The purpose of this design is to extend the loading capacitive range with low power consumption and wide bandwidth by comparing with the state-of-the-arts [8–10]. Some key parameters of the proposed amplifier circuit are shown in Table 2, and the relevant transistor sizes can be found in Table 3.

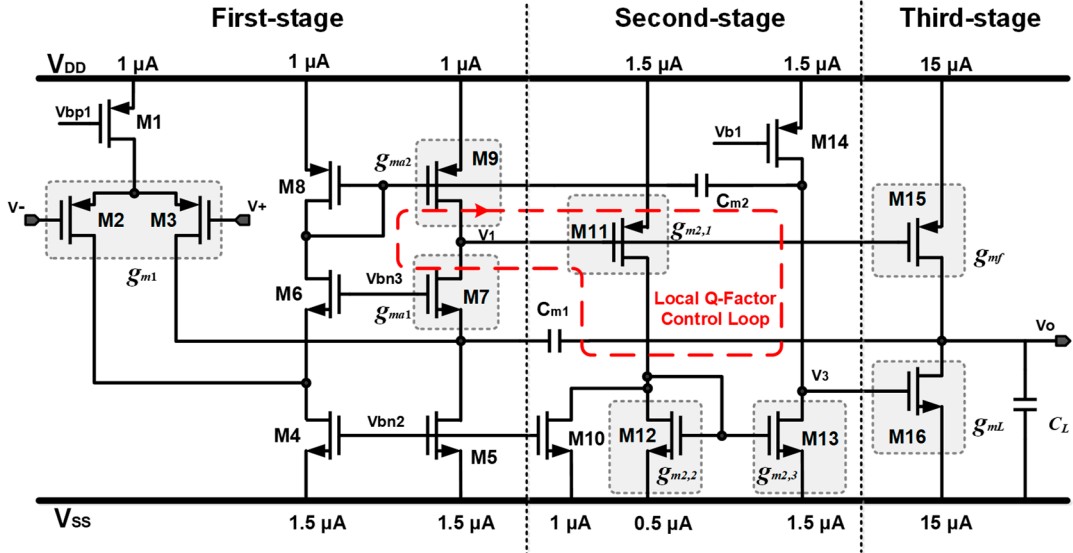

**Figure 8.** The simplified schematic of the proposed three-stage CLQC amplifier.

**Table 2.** Circuit parameters of the proposed amplifier.

| $g_{m1}$ | $g_{m2}$ | $g_{mL}$ | $g_{ma1}$ | $g_{ma2}$ | $g_{mf}$ | $C_{m1}$ | $C_{m2}$ |
|---|---|---|---|---|---|---|---|
| 7.5 µS | 32 µS | 580 µS | 18 µS | 14 µS | 560 µS | 1 pF | 0.05 pF |

**Table 3.** Transistor sizes.

| Transistor | $M_1$ | $M_{2,3}$ | $M_{4,5}$ | $M_{6,7}$ | $M_{8,9}$ | $M_{10}$ | $M_{11}$ | $M_{12}$ | $M_{13}$ | $M_{14}$ | $M_{15}$ | $M_{16}$ |
|---|---|---|---|---|---|---|---|---|---|---|---|---|
| W/L (µm) | 1/2 | 4/2 | 0.4/3 | 1/0.8 | 2/1 | 0.4/3 | 0.6/0.13 | 0.6/0.13 | 2/0.64 | 3/1 | 2/0.64 | 0.64/0.32 |
| Multiple | 2 | 2 | 3 | 1 | 1 | 2 | 1 | 3 | 1 | 1 | 10 | 5 |

## 5. Simulation Results

The proposed three-stage amplifier is verified in 0.13-µm CMOS technology. All transistors are implemented by standard threshold voltage devices. The active region occupies 0.0036-mm$^2$ (30 × 120 µm) die area and the chip layout is depicted in Figure 9. The total on-chip capacitance $C_t$ is 1.05 pF ($C_{m1}$ = 1.0 pF and $C_{m2}$ = 0.05 pF). Figure 10 shows a series of Bode plots from simulations with various values of $C_L$ ranging from 4 pF to 1.5 nF for the proposed scheme. When $C_L$ is equal to 1.5 nF, the corresponding UGF and PM are 0.88 MHz and 42.3°, respectively. When $C_L$ is reduced to 0.5 nF, both UGF and PM are increased to 0.9 MHz and 62.5°, respectively. When $C_L$ is further dropped to 150 pF, the UGF is extended to 0.92 MHz with PM = 89.6°. Additionally, when $C_L$ is as small as 4 pF, the corresponding UGF and PM are increased to 0.97 MHz and 95.0°. As indicated in Figure 10, there is no obvious frequency peaking for the proposed design within the load capacitance range from 4 pF to 1.5 nF. It is worth mentioning that the worst-case PM of 42.3° is to demonstrate the maximum value of $C_L$ (1.5 nF) that the proposed amplifier can support to closely meet the empirical minimum PM of 45° under 24.0-µW power consumption. With larger biasing current in the last gain stage, the output capacitance value or PM can be increased accordingly.

Table 4 summarizes the simulated PM, GM and UGFs under different corners and temperatures when $C_L$ is 1 nF. At 27 °C, it can be found that the maximum PM deviation is around 5° for different corners. They are similar in the cases of –40 and 125 °C. Under five-corner process variation and –40 to 125 °C temperature range, the simulated minimum phase margin is 40.3°, and the minimal gain margin is 13.2 dB, which indicates stable operation for the proposed design is achieved.

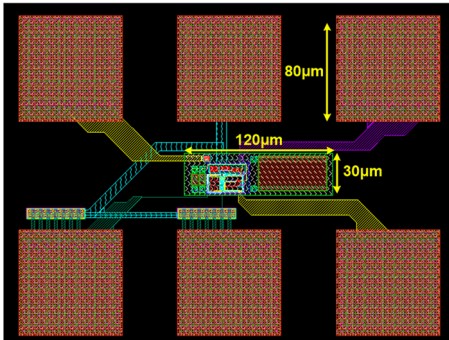

**Figure 9.** Layout of the proposed amplifier circuit.

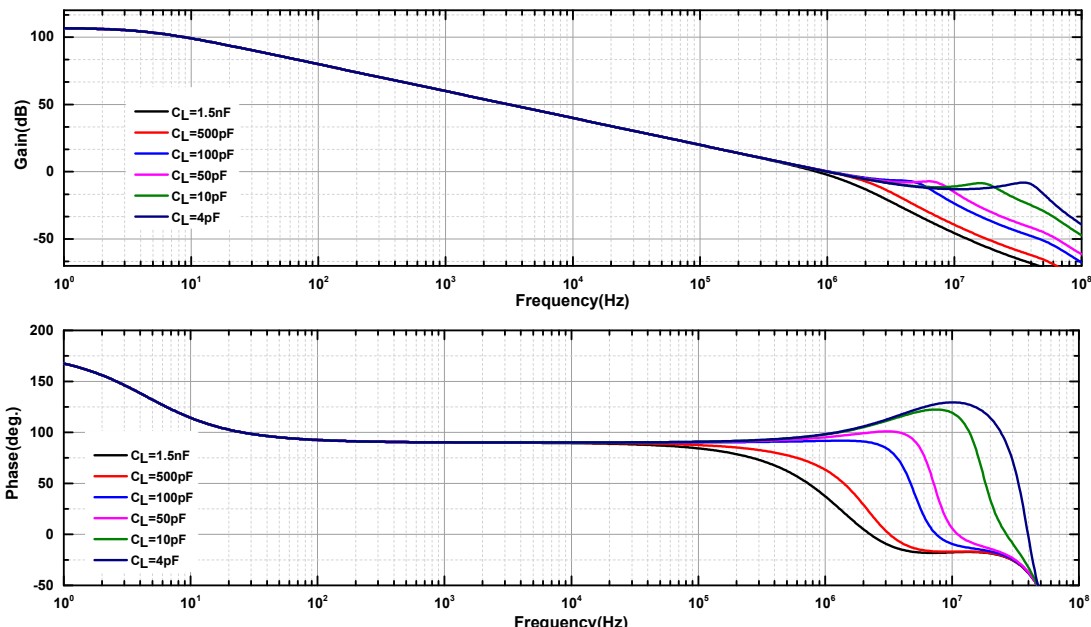

**Figure 10.** AC simulation results with various values of $C_L$ ranging from 4 pF to 1.5 nF.

**Table 4.** Simulation results at different temperatures and process corners.

| Corner * | TT | FF | SS | SF | FS |
|---|---|---|---|---|---|
| | | T = −40 °C | | | |
| UGF (MHz) | 0.80 | 0.75 | 0.92 | 0.95 | 0.90 |
| PM (°) | 46.0 | 41.4 | 43.2 | 46.0 | 49.8 |
| GM (dB) | 14.0 | 14.9 | 13.8 | 13.9 | 13.7 |
| | | T = 27 °C | | | |
| UGF (MHz) | 0.88 | 0.77 | 0.80 | 0.93 | 0.89 |
| PM (°) | 45.3 | 40.2 | 46.4 | 45.5 | 45.6 |
| GM (dB) | 13.5 | 15.8 | 14.2 | 13.5 | 13.2 |
| | | T = 125 °C | | | |
| UGF (MHz) | 0.78 | 0.72 | 0.70 | 0.72 | 0.75 |
| PM (°) | 44.2 | 42.3 | 40.2 | 43.0 | 45.4 |
| GM (dB) | 14.1 | 15.1 | 14.5 | 14.6 | 13.8 |

* TT means both NMOS and PMOS are in typical condition; FF means both NMOS and PMOS are in fast condition; SS means both NMOS and PMOS are in slow condition; SF means NMOS is in slow condition and PMOS is in fast condition; and FS means NMOS is in fast condition and PMOS is in slow condition.

Figure 11 shows the AC response of the proposed amplifier with different loading capacitance from 4 pF to 1.5 nF under ±0.2-V supply voltage variations. On one hand, thanks to the cascode structure of

the first gain stage which contributes the majority of the overall voltage gain of the amplifier, the DC voltage gain only varies between 92 and 107 dB. On the other hand, with the proposed frequency compensation scheme, the non-dominant pole locations are almost free from the change with the input voltage variations. Therefore, a very small phase difference can be spotted in the series of phase plots with the same loading capacitance. In general, this figure illustrates the three-stage amplifier with proposed frequency scheme can provide robust operation under supply voltage variations.

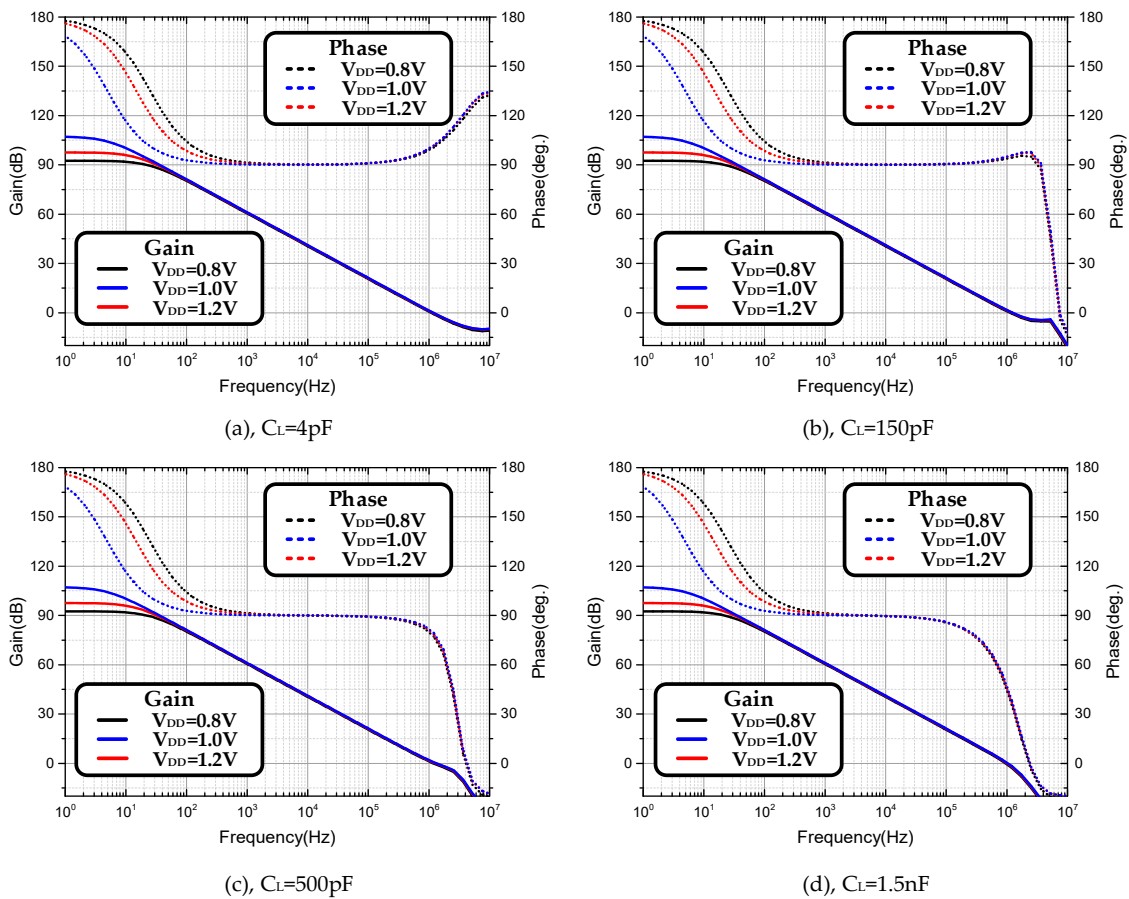

**Figure 11.** AC simulation results (**a**–**d**) with $C_L$ (4 pF to 1.5 nF) when $V_{DD}$ varies from 0.8 to 1.2 V.

The simulated transient responses of the proposed amplifier in unity-gain configuration with a 500 mV step for $C_L$ to be 4 pF, 150 pF, 500 pF and 1.5 nF are shown in Figure 12a–d. With 4-pF loading capacitance, the average slew rate is 0.58 V/μs, and the average 1% settling time is 0.15 μs. When the load capacitance is 150 pF, the relevant SR and settling time are 0.62 V/μs and 0.15 μs, respectively. If a 500-pF capacitive load is applied at the output, the average SR and average 1% settling time are 0.57 V/μs and 0.5 μs, respectively. When $C_L$ is increased to 1.5 nF, the corresponding SR and average 1% settling time are 0.41 V/μs and 1.0 μs, respectively.

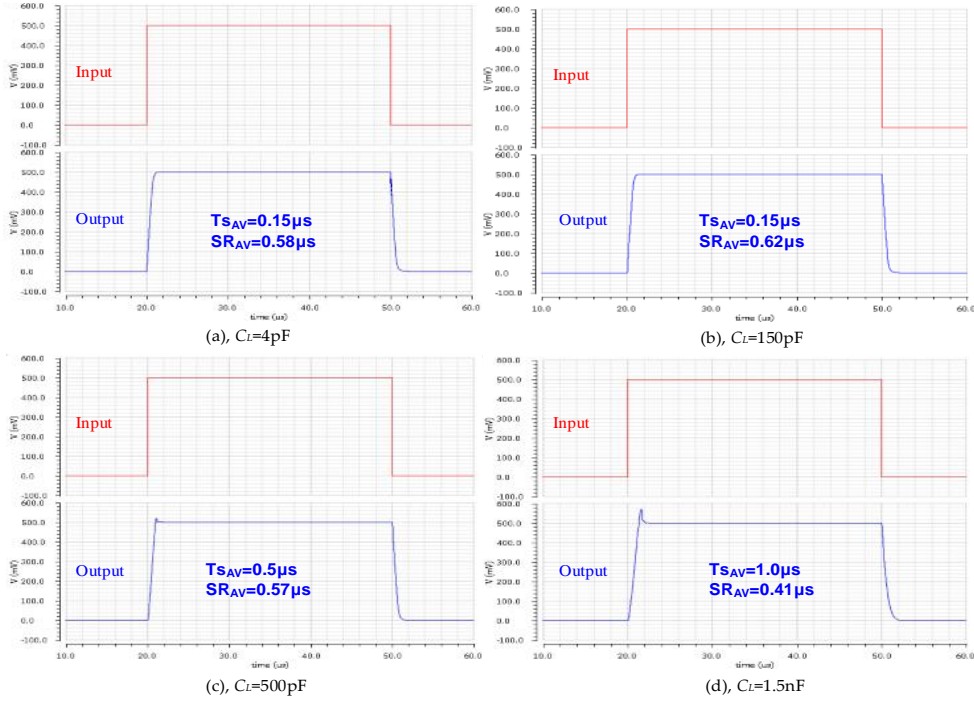

**Figure 12.** Simulated 500 mV step responses at (**a**) $C_L$ = 4 pF (**b**) $C_L$ = 150 pF (**c**) $C_L$ = 500 pF and (**d**) $C_L$ = 1.5 nF.

The simulation results of power-supply rejection ratio (PSRR) and common-mode rejection ratio (CMRR) with open-loop response are depicted in Figure 13a,b. The PSRR and CMRR are around 95 and 105 dB at 1 kHz, respectively.

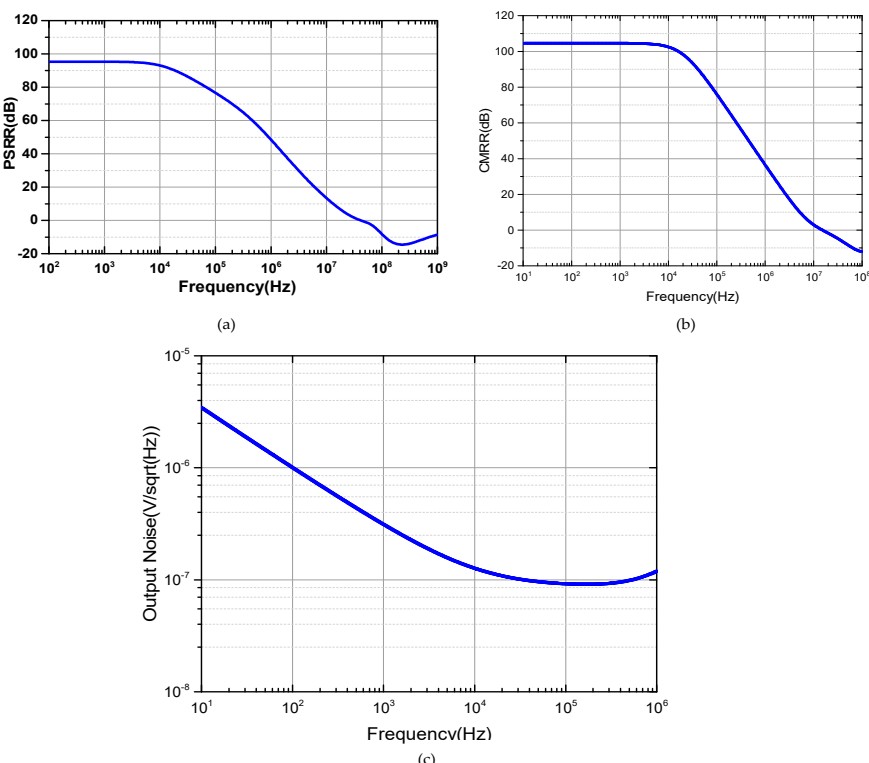

**Figure 13.** Simulated results of the proposed amplifier (**a**) PSRR (**b**) CMRR and (**c**) output noise density.

The simulated output noise density spectrum of the proposed amplifier which is configured as a unity-gain buffer is shown in Figure 13c. The corner frequency of the 1/f noise is close to 8 kHz and the white noise amplitude is about 94 nV/$\sqrt{\text{Hz}}$ at 100 kHz.

Table 5 summarizes the performance of the proposed CLQC amplifier with the recent high-gain (>100 dB) amplifiers with the driving capability more than 10× $C_L$. It can be seen from this table that the proposed three-stage amplifier achieves the largest drivability (375x) over other designs. The proposed design can provide a stable operation with the load capacitance ranging from 4 pF to 1.5 nF, which is very suitable for analog signal processing applications requiring high gain and high bandwidth.

**Table 5.** Performance summary and comparison with recent works.

| Specifications | EL'15 [29] | | | TCAS-I'16 [30] | | | This Work | | |
|---|---|---|---|---|---|---|---|---|---|
| Drivability | 10x | | | 150x | | | 375x | | |
| Load $C_L$ | 150 pF | 1 nF | 1.5 nF | 100 pF | 1.5 nF | 15 nF | 4 pF | 150 pF | 500 pF | 1.5 nF |
| Technology | 0.18-μm CMOS | | | 0.18-μm CMOS | | | 0.13-μm CMOS | | |
| Chip Area* (mm$^2$) | 0.0045 | | | 0.0021 | | | 0.0036 | | |
| DC Gain | >100 dB | | | 100 dB | | | >100 dB | | |
| UGF (MHz) | 1.60 | 1.13 | 0.89 | 1.66 | 0.12 | 0.01 | 0.97 | 0.92 | 0.90 | 0.88 |
| PM (°) | 76.7 | 56.2 | 50.0 | 69 | 87 | 85 | 95.0 | 89.6 | 62.5 | 42.3 |
| Power | 15.8 μW @ 1.2 V | | | 7.4 μW @ 1.1 V | | | 24.0 μW @ 1.0 V | | |
| On-chip Cap. | 1.0 pF | | | 0 | | | 1.05 pF | | |
| On-chip Res. | 125 kΩ | | | 17.7 kΩ | | | 0 | | |
| Average SR (V/μs) | 0.76 | 0.41 | 0.28 | 8.67 | 5.87 | 1.1 | 0.58 | 0.62 | 0.57 | 0.41 |
| Average 1% Ts (μs) | 2.16 | 3.87 | 5.34 | 1.2 | 4.3 | 2.4 | 0.15 | 0.15 | 0.5 | 1.0 |

## 6. Conclusions

In this paper, a low-power (24.0-μW) three-stage CMOS amplifier with 375x capacitive load ($C_L$) drivability range is presented. Combining cascode and Miller compensation, the complex-pole frequency $\omega_o$ is extended effectively which enables higher GBW of the proposed amplifier. Pushing the complex-pole to higher frequency while lowering its $Q$-factor are a contradiction as studied in prior designs. Thanks to the proposed CLQC technique, the $Q$-factor of the complex poles is restricted to an appropriate range while the complex-pole frequency $\omega_o$ is maintained according to different $C_L$ applied at the output. Therefore, an optimized tradeoff between complex-pole frequency $\omega_o$ and the $Q$-factor is achieved. The proposed amplifier was verified by 0.13-μm CMOS technology, and the simulation results show at least 0.88-MHz GBW and 0.41 V/μs average are achieved under 4-pF-to-1.5-nF $C_L$, and the on-chip compensative capacitance is only 1.05 pF.

**Author Contributions:** Conceptualization, Q.C. and J.G.; Data curation, Q.C. and W.L.; Funding acquisition, X.T. and J.G.; Investigation, Q.C., W.L. and J.G.; Methodology, Q.C. and J.G.; Project administration, X.T. and J.G.; Supervision, J.G.; Writing—original draft, Q.C.; Writing—review & editing, W.L., X.T. and J.G.

**Funding:** This research was funded by National Natural Science Foundation of China, grant number 61874143, and Shenzhen Research and Development Funds for Science and Technology, grant number JCYJ20180508152019687.

**Conflicts of Interest:** The authors declare no conflict of interest.

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
