# Peer review of "Design and Analysis of Three-Stage Amplifier for Driving pF-to-nF Capacitive Load Based on Local Q-Factor Control and Cascode Miller Compensation Techniques"

_electronics, doi:10.3390/electronics8050572_

Round 1

Reviewer 1 Report

The paper analytically describes the proposed three-stage amplifier designed for low-voltage low-power applications where high capacitive load driving is required. Actually, the real case application (e.g., mobile applications, IOT, sensor interface) is completely missing and the paper risks to represent theoretical exercise with no impact on the development of a research or commercial device. I suppose that the authors effort has been addressed to solve the specific issue of high load capacitance and frequency stability within the design of a more complex system or for a specific research topic, but they are not mentioned in the manuscript. A reader question could be "why is this architecture useful for?".

The authors refer to previous work to show the advancement of the project during the years, but the improvement from ISCAS 2017 is not really visible. Large analytical part is added but simulations results are similar and measurements on silicon are missing.  

Here below, some comments on the single sections of the manuscript.

- Introduction and comparison with state-of-the-art is well described. Description of real applications where the proposed circuit can be useful is missing and related motivations should be stressed.

- Section 2 could be slightly shortened, keeping figures and moving important formulas in a single summarizing table. Anyway, this section is well described.

- Section 3 should be strongly changed and revised since it includes sentences totally copied from ISCAS paper 2017. For instance, figure 4 is the same of figure 1 in ISCAS paper. Please, make a reference to previous work but do not copy whole sections or figures. Mathematic described in section 3.3 is extensively reported but the link to physical implementation (section 4) seems to be weak. How do the authors size transistors (and their gm) and capacitances in the circuit basing on the formulas described in section 3?

- Section 4 should be improved with value of bias currents. This section is important and should be make stronger with additional details.

- Section 5 is poor of novelty. Main simulation results are already present in ISCAS 2017. CMRR, PSRR, noise analysis (e.g., input referred noise value) are completely missing. Moreover, corner simulations include process and temperature variations, but not supply voltage variations. Please, add it or justify why it is not required for specific application. In my opinion, the reader could have the suspect of low performances for voltage supply variations.

Author Response

Dear Editor-in-Chief, Associate Editor, and Reviewers,

The authors would like to express our sincere appreciation to the Editor, Associate Editor and Reviewers for their time and efforts in providing valuable comments and suggestions. The reviewers’ comments have been carefully studied, and we have revised our manuscript following the suggestions from the reviewers. The changes are highlighted in red in the revised manuscript.

Should there be any problem about this submission, please feel free to contact me. Thank you.

Sincerely,

Jianping Guo

Associate Professor

School of Electronics and Information Engineering

Sun Yat-sen University, Guangzhou, China

Tel: +86 8411 4462

Reviewer 2 Report

This article considers the analysis and design of a three-stage operational amplifier (op-amp) that can operate over a wide range of output load capacitance conditions. The paper is well structured, logically ordered and clear to the reader as to what is being discussed. The two key aspects of the work which are interesting and useful are (1) the design and analysis of a three-stage op-amp, and (2) the wide range of output load capacitance values. The design and analysis are suitably discussed and useful simulation results are provided. The work discussed here seems complete and covers many aspects of more complex op-amp architectures and design. 

Possible considerations are:

Is it "Analysis and Design" or "Design and Analysis" in the title?

It would be beneficial to review the paper for consistency for the presentation of values and units. For example, should the style be a space between the number and the units. In some places, there is a space. In others there is no space and in some places there is a "-" used.

In the Abstract, the authors discuss the phase margin (42.3 degrees). It would be useful to include in the paper, some additional discussion into this number as the op-amp is typically designed with a minimum PM of 45 degrees and preferably around 60 degrees. 

Page 1, line 34. What exactly is meant by "further degraded"?

Page 2, line 65 - can the "small-signal figure-of-merit" be explained in more detail.

Page 4, line 122 - is it "unit" or "unity"?

Page 6, line 181 - Should the "M" in "Miller" be a capital letter?

Author Response

Dear Editor-in-Chief, Associate Editor, and Reviewers,

The authors would like to express our sincere appreciation to the Editor, Associate Editor and Reviewers for their time and efforts in providing valuable comments and suggestions. The reviewers’ comments have been carefully studied, and we have revised our manuscript following the suggestions from the reviewers. The changes are highlighted in blue in the current revised manuscript.

Should there be any problem about this submission, please feel free to contact me. Thank you.

Sincerely,

Jianping Guo

Associate Professor

School of Electronics and Information Engineering

Sun Yat-sen University, Guangzhou, China

Tel: +86 8411 4462

Round 2

Reviewer 1 Report

The paper has been strongly improved and the simulation results have been completed with the suggested analysis. The advantage of the presented circuit topology and the design strategy have been reported in detail to stress the impact to real application cases.

Minor changes are still required to refine the manuscript and some suggestions are addressed to the authors:

          Line 72-73)  “loading capacitance may vary from several pico-Farads to several or more than ten nano-Farads in some applications” should be simplified to “load capacitance can change in the range of pF-nF depending on application, such as headphone, ….”

               Line 82-83) if possible, change “very few of them …. small chip area” to “it is hard to find amplifier designs able to combine the possibility to drive capacitive load in the pF and nF range with low quiescent power and small active area”

               Line 83-84) remove “since the closed…. heavy CL conditions” because you have already mentioned it at line 79-80.

               Line 91-92)  “Large GBW” and “reasonable power consumption” are ambiguous term and could be subjective. What is Large and Reasonable? For instance, it depends on applications. Specify numbers coming from simulations, so that the reader can understand if it is large or reasonable for his case study.

               Line 121-122) The sentence “Eventually, the complex pair…. reduced to 0.5.” could be merged with the sentence at line 126-127:  “Eventually, they will split into two real poles and Q will be reduced to 0.5.”

               Line 168) remove “will”

               Figure 5 could be enlarged for a better comprehension. It is the small-signal model related to the block diagram of figure 4. Similar figure might be added to represent the small-signal model of the transistor level of the proposed circuit (shown in figure 8)

               Line 205)  R1,R2,R3 are not defined. There might be a typing error.

               Figure 6 could be enlarged and some reference should be added to understand the pole movement with CL increase (arrows or notations, as in figure 3b)

               Sentences at Line 277 and 278 could be merged.

               Line 294) what do you refer to with the expression “frequency response could be optimized”?  Do you refer to the absence of gain peak at high frequence? Do you often use the expression “frequency optimization” but it is ambiguous….

               Line 301) Please, check formula of transconductance gm2.

               In Figure 11, dotted lines representing phase is not easily visible. Please use another size for such lines.

               If possible, expand last section regarding conclusions

Author Response

(The authors gave the same response as above.)
